# Calprotectin in Patients with Rheumatic Immunomediated Adverse Effects Induced by Checkpoints Inhibitors

**DOI:** 10.3390/cancers15112984

**Published:** 2023-05-30

**Authors:** Beatriz Frade-Sosa, Chafik Alejandro Chacur, Josep M. Augé, Andrés Ponce, Juan C. Sarmiento-Monroy, Ana Belén Azuaga, Nuria Sapena, Julio Ramírez, Virginia Ruiz-Esquide, Rosa Morlà, Sandra Farietta, Patricia Corzo, Juan D. Cañete, Raimon Sanmartí, José A. Gómez-Puerta

**Affiliations:** 1Department of Rheumatology, Hospital Clinic of Barcelona, Institut d’Investigacions Biomèdiques August Pi i Sunyer (IDIBAPS), 08036 Barcelona, Spain; 2Department of Biochemistry and Molecular Genetics (CDB), Hospital Clinic of Barcelona, 08036 Barcelona, Spain

**Keywords:** calprotectin, immune checkpoint inhibitors, immune-related adverse events, rheumatoid arthritis

## Abstract

**Simple Summary:**

The search for serum biomarkers of inflammatory activity in patients with immune-mediated diseases is an area of interest and has been the subject of multiple investigations in recent years. Calprotectin is a biomarker that can be used to identify inflammation and tissue damage and provide information on the extent and location of the damage. In patients with rheumatoid arthritis, the measurement of calprotectin levels is a sensitive biomarker for monitoring activity and a prognostic factor for the disease. It is well known that immune checkpoint inhibitors can cause immune-related adverse effects due to their mechanism of action. Rheumatic adverse effects have been reported and are increasingly recognized. Calprotectin determination may be useful in the evaluation of these patients and in monitoring disease activity. There are no reports of calprotectin in patients with immune-related rheumatic adverse effects.

**Abstract:**

Background: this is an exploratory study to evaluate calprotectin serum levels in patients with rheumatic immune-related adverse events (irAEs) induced by immune checkpoint inhibitor (ICI) treatment. Methods: this is a retrospective observational study including patients with irAEs rheumatic syndromes. We compared the calprotectin levels to those in a control group of patients with RA and with a control group of healthy individuals. Additionally, we included a control group of patients treated with ICI but without irAEs to check calprotectin levels. We also analysed the performance of calprotectin for the identification of active rheumatic disease using receiver operating characteristic curves (ROC). Results: 18 patients with rheumatic irAEs were compared to a control group of 128 RA patients and another group of 29 healthy donors. The mean calprotectin level in the irAE group was 5.15 μg/mL, which was higher than the levels in both the RA group (3.19 μg/mL) and the healthy group (3.81 μg/mL) (cut-off 2 μg/mL). Additionally, 8 oncology patients without irAEs were included. In this group, calprotectin levels were similar to those of the healthy controls. In patients with active inflammation, the calprotectin levels in the irAE group were significantly higher (8.43 μg/mL) compared to the RA group (3.94 μg/mL). ROC curve analysis showed that calprotectin had a very good discriminatory capacity to identify inflammatory activity in patients with rheumatic irAEs (AUC of 0.864). Conclusions: the results suggest that calprotectin may serve as a marker of inflammatory activity in patients with rheumatic irAEs induced by treatment with ICIs.

## 1. Introduction

The use of immune checkpoint inhibitors (ICIs) has revolutionized oncologic treatment and has improved survival rates for patients with various types of cancer. There are several types of ICIs, including those that target cytotoxic T-lymphocyte-associated antigen-4 (CTLA-4) and the programmed death-1 (PD-1)/programmed death-ligand 1 (PD-L1) pathway [1].

It is well known that ICIs can cause side effects; these are known as immune-related adverse events (irAEs) due to their mechanism of action. Rheumatic irAEs, including ICI-induced arthritis [2,3,4,5,6], a clinical syndrome resembling polymyalgia rheumatica, myositis [7,8,9], fasciitis [10], vasculitis [11], and sarcoidosis-like [12] or systemic lupus erythematosus [13], among others, have been reported and are being recognized more frequently.

Calprotectin (S100A9/S100A8) is a heterodimeric complex, a member of the S100 protein family, which is released by cells of the innate immune system, such as neutrophils and monocytes, during inflammation; its levels in the blood can be measured to evaluate the presence and severity of inflammation. It has proinflammatory activities and acts as endogenous-associated molecular patterns via Toll-like receptor activation [14].

Studies have shown that calprotectin levels are minimal in blood and stool samples from healthy populations compared to patients with inflammatory disorders [15,16,17,18]. Calprotectin is a sensitive marker of inflammation in various conditions [19]. Increased calprotectin expression has also been found in patients with rheumatic diseases [20]. In rheumatoid arthritis (RA) patients, serum and plasma calprotectin levels are sensitive markers of inflammation [21,22] and have been associated with radiographic damage [23]. In addition, calprotectin can be a biomarker of clinical responses to antirheumatic drugs [24] and a predictive factor for disease relapse [25].

Based on this evidence, serum calprotectin determination has recently been routinely introduced in rheumatology clinical practice in our hospital as a new biomarker of inflammation [26]. In 2022, we conducted a preliminary study to assess the role of a new biomarker. We analysed all calprotectin tests conducted in our hospital in the previous year and found 2655 tests with a mean result of 2.8 ± 3.1 µg/mL. Of these, 48 (1.8%) showed elevated levels (≥8.9 µg/mL, more than 2 standard deviations of the cut-off (2 µg/mL)) and corresponded to 33 patients. Of these patients, 60% had RA, while others had conditions such as spondylarthritis (*n* = 3), systemic lupus erythematosus (*n* = 2), palindromic rheumatism (*n* = 2), systemic sclerosis (*n* = 1), and adult Still’s disease (*n* = 1). Moreover, 2 patients (6%) had rheumatic irAEs due to immunotherapy. None of the patients had active infections at the time of testing.

We have previously described the different clinical patterns of rheumatic irAEs and their rheumatic and oncologic outcomes [6] and noted that ICI-induced arthritis patients present inflammatory patterns on imaging studies similar to conventional inflammatory arthritis [9]. There are no reports exploring the role of serum calprotectin in patients with rheumatic irAEs. This exploratory study aimed to evaluate serum calprotectin levels in a group of patients with rheumatic irAEs focused on patients with induced inflammatory arthritis and to compare them with a group of patients with RA, a group of patients under ICI treatment without irAEs and with a control group of healthy people. We hypothesize that calprotectin is also a marker of inflammatory activity in oncologic patients with rheumatic irAEs and can identify patients with active inflammatory processes. However, we posit that levels will be higher in oncologic patients than in patients with active RA given the inflammatory condition of their underlying disease.

## 2. Materials and Methods

### 2.1. Design and Study Population

We conducted a retrospective observational study including all adult patients referred to the Rheumatology Department of our centre due to the onset of rheumatic syndromes related to ICI treatment who underwent the determination of serum calprotectin. Data collected included demographic features, history of previous rheumatic diseases, ICI indication and type, and disease manifestations at irAE onset. We classified clinical syndromes according to 4 different categories: (a) inflammatory arthralgia, (b) RA-like, (c) oligo/polyarthritis with or without enthesitis and tenosynovitis, and (d) polymyalgia rheumatica (PMR), as in patients with inflammatory proximal muscle pain with or without arthritis. Patients with non-inflammatory arthralgia were excluded. We identified those patients in whom calprotectin determination was routinely performed as a biomarker of inflammatory activity.

As the control group, we used a cohort of RA patients (ACR/EULAR 2010 [26]) from our arthritis unit who were admitted consecutively between July 2020 and July 2021. Patients were included regardless of their disease activity status, previous use of disease-modifying anti-rheumatic drugs (DMARDs) (including biological therapies or JAK inhibitors (JAKi)), and concomitant treatment (methotrexate or others). Patients who, at the study visit, presented signs of an active infection or other clinical conditions that, in the opinion of the investigator, could modify the results of calprotectin determinations were excluded. All patients underwent clinical assessment, including 28 swollen and tender joint counts (28SJC and 28TJC) and physician and patient global assessment (PhGA and PGA) with visual analogue scales (0–10). Disease activity indices were subsequently calculated. These included the disease activity score (DAS28), simplified disease activity index (SDAI), and clinical disease activity index (CDAI). We used the DAS28 to classify patients according to their degree of disease activity, and a subgroup of patients with active disease (DAS28 > 2.6) was identified. We also included a second control group of healthy volunteers, selected from hospital workers with no medical history of interest. To validate our observations, we also determined the levels of calprotectin in a group of cancer patients undergoing treatment with ICI without any irEAs undergoing follow-up observations by the oncology service of our hospital; these patients were attending their control visit and voluntarily agreed to participate in the study as a control group.

### 2.2. Assessment of Calprotectin

Calprotectin serum levels were determined using a DiaSorin Liaison^®^ Calprotectin assay (DiaSorin, Saluggia, Italy) [26]. We compared the levels of serum calprotectin determined in each of the groups. We also compared serum calprotectin levels between subgroups of patients with active inflammatory disease at the time of the determination (ICI-induced arthritis vs. RA). The cut-off point in the technique was set at 2 μg/mL [26].

### 2.3. Statistical Analysis

Descriptive statistics were reported as the mean ± standard deviation for continuous variables and frequency and proportions for categorical variables. Differences between groups’ outcomes were compared using Mann–Whitney U-tests. Comparisons between groups were initially conducted in all patients and therefore a subgroup analysis was done including only patients with active rheumatic disease.

Correlation analysis (Spearman’s correlation coefficient) was used to assess the association between serum calprotectin and classic acute phase reactants (CRP and ESR). The performance of calprotectin for the diagnosis of active rheumatic disease (inflammatory activity: yes/no) in the irAEs group was analysed using receiver-operating characteristic curves (ROC). The ROC curves made it possible to calculate the area under the curve (AUC) as a measure of the overall discriminative power. The performance of ESR and CRP was also studied, and the discriminative power of the three determinations combined was compared with the isolated determinations for the diagnosis of active rheumatic disease.

The statistical analysis was conducted using SPSS software (IBM SPSS version 27.0, Inc., Chicago, IL, USA) and graphics were produced with GraphPad Prism version 7.0 (GraphPad Software, San Diego, CA, USA). All patients gave written informed consent. The study was approved by the Hospital Clinic Institutional Review Board (HCB/2021/0901).

## 3. Results

### 3.1. Patients

Eighteen patients undergoing treatment with ICIs who had been referred to the Rheumatology Department with rheumatic irAEs were included. The mean age was 61 years and 44% were female. Of all the patients referred, six had active arthritis at the time of the calprotectin assessment. Table 1 summarizes the characteristics of the patients.

The control group consisted of 128 patients with RA (90% female, mean age 55.9 years old, 85.6% were seropositive (anti-rheumatoid factor (RF) and/or anti-citrullinated protein antibodies (ACPA)). Of these patients, 90 were undergoing biological treatment (57 with IL-6 inhibitors, 30 with TNF inhibitors, and 3 with rituximab), 28 were undergoing JAK inhibitor treatment, and 11 were being treated with conventional DMARDs. The patients exhibited different degrees of disease activity (mean DAS28 3.38 (1.60)). In total, 60% of the patients had active disease (DAS28 > 2.6): 19 patients had low disease activity (DAS 28 2.6–3.2), 43 had moderate disease activity (DAS28 3.2–5.1), and 17 had high disease activity (activity (DAS28 ≥ 5.1). Table 2 summarizes the characteristics of the RA group. A second control group of 29 healthy donors was also included. The mean age was 47.23 (10.05) years and 73.3% of them were female.

### 3.2. Serum Calprotectin Levels

The mean serum calprotectin levels in patients with rheumatic irAEs were 5.15 ± 4.96 μg/mL. In the control group of RA patients, the mean serum calprotectin was 3.19 ± 3.60 μg/mL. This difference did not reach statistical significance (*p* value = 0.8). In the healthy donor group, the mean serum calprotectin was 3.81 ± 1.86 μg/mL, significantly lower than that of the main group (*p* value = 0.006). Furthermore, we performed a determination of serum calprotectin in eight patients receiving ICI treatment but without irAEs who volunteered for the study. The mean levels of calprotectin in these patients was 2.41 ± 2.05 μg/mL (Figure 1). We found that these levels were significantly lower than those of patients with irAEs (*p* value 0.49) and without significant differences compared to the healthy controls (*p* value 0.56).

We also compared the acute-phase reactants. We found that CRP and ESR showed significant differences between the groups (Table 3).

When classifying patients with ICI as patients with active inflammatory activity (*n* = 6) versus patients without inflammatory activity (*n* = 12) at the time of the calprotectin assessment, we observed that serum calprotectin levels were higher in the group of active patients (8.43 μg/mL (7.2) vs. 3.53 μg/mL (2.47)). Our findings showed a statistical trend toward higher levels of serum calprotectin in patients with inflammatory activity (*p* < 0.05). Similarly, when comparing serum calprotectin levels in the group of patients with active RA (DAS28 > 2.6) versus patients in remission, we also found significant differences in serum calprotectin levels in active RA patients (3.94 μg/mL (4.25) vs. 2.03 μg/mL (1.72), *p* < 0.05)).

### 3.3. Serum Calprotectin Levels in Patients with Active Arthritis

We performed a sub-analysis to compare the serum levels of calprotectin in patients with active arthritis in both groups (6 patients in the study group and 79 in the RA control group). Although both groups had elevated serum calprotectin levels, we found significantly higher levels of serum calprotectin in the group of patients treated with ICI (8.43 (7.2) μg/mL vs. 3.94 (4.25), *p* = 0.013)) (Figure 2). No significant differences were observed with CRP or ESR (Table 4).

### 3.4. Discriminatory Capacity of Serum Calprotectin to Identify Inflammatory Activity in Patients with irAEs

We studied the discriminatory capacity of serum calprotectin to identify patients with active inflammatory activity in the group of patients with irAEs and compared it with the acute-phase reactants. We found that ESR and CRP had moderate discriminatory capacity to identify patients with inflammatory activity, with AUC values of 0.608 and 0.697, respectively (Figure 3a,b). However, serum calprotectin had a very good discriminatory capacity with an AUC of 0.864 (Figure 3c).

The performance of the three biomarkers combined (calprotectin, ESR, and CRP) did not increase the sensitivity of the single calprotectin determination (Figure 4).

The analysis of the correlation between blood biomarkers showed a low correlation between calprotectin and ESR (r = 0.283, *p* < 0.01) and a moderate correlation between calprotectin and CRP. The correlation between CRP and ESR was also moderate (r = 0. 434, *p* < 0.01) (Table 5).

## 4. Discussion

ICIs are widely used in the treatment of various types of cancer. Although they have significant clinical benefits, these therapies are associated with a wide spectrum of irAEs. Currently, it is difficult to generalize the evidence in the existing literature on risk factors or biomarkers for the entire class of ICIs because the studies are either disease-specific or ICI-agent-specific studies.

The optimal biomarker to predict the risk of irAEs remains undefined. Several clinical parameters and biomarkers have been associated with greater toxicity, but none of them has yet been prospectively validated. They include the presence of pre-existing autoimmune disease, chronic smokers, sex and body mass index, tumour response to ICI, circulating cytokines, inherited genetic variants, and specific gut microbiome composition [41,42]. Additionally, patients with significant kidney (stage IV–V), cardiac, and lung disease are at a higher risk of respective organ-specific irAEs [42]. In terms of pre-existing autoimmune diseases, plenty of evidence also suggests that this group of patients has a greater risk of developing irAEs [43,44]. It is interesting to highlight the risk of gastrointestinal adverse events in patients with a prior history of IBD [44,45]. In the present study, we found higher levels of conventional biomarkers such as ESR and CRP in patients with rheumatic irAEs in comparison with RA patients and healthy controls, as well as higher levels of serum calprotectin, despite not reaching statistical significance. We also observed that the serum calprotectin in patients with irAEs was higher than in ICI-treated patients without irAEs. Despite the small sample, this results reinforces the idea that serum calprotectin may be a biomarker for the identification of arthritis induced by ICI.

Interestingly, serum levels of calprotectin were higher in patients with active disease in patients with rheumatic irAEs vs. RA. The classical acute-phase reactants (CRP and ESR) did not show statistically significant differences between the groups (Table 4). It seems that serum calprotectin might even more accurately identify patients with inflammatory activity, as shown in the ROC curve analysis.

Another notable result is that the serum calprotectin levels in healthy controls and the RA group were similar, as well as the classic acute-phase reactants. This may be explained in part by the fact that more than half of the RA patients were in remission or exhibited low activity. These findings should be interpreted carefully due to our sample size and the difficulty of interpreting serum biomarkers in patients with underlying neoplasia. The evaluation of biomarkers in patients with cancer is a complex issue and involves the interpretation of the results in the context of the patient’s overall clinical presentation, type of treatment, and cancer stage, among other factors. Biomarkers can be used to help guide treatment decisions and to monitor responses to therapy.

Calprotectin expression is not homogeneous in all types of cancers and varies according to the cell and tissue of origin; for instance, it is reported to be downregulated in cells from head and neck squamous cell carcinoma [46]. Serum calprotectin and faecal calprotectin have been used as diagnostic biomarkers in patients with colorectal cancer [47,48].

To the best of our knowledge, this is the first study to evaluate the clinical value of serum calprotectin in patients undergoing ICI therapy and irAEs. Our study has some limitations including the small sample size of patients with rheumatic irAEs and the cross-sectional design, which precluded us from assessing changes in calprotectin concentrations over time or after treatment. An additional limitation is the lack of information about the cut-off value of serum calprotectin in patients with malignancies and/or in patients with other irAEs. Finally, the monocentric nature of our study may limit the external validity of the results.

## 5. Conclusions

The diagnosis and follow-up of patients with irAEs is a novel area for clinicians dealing with patients undergoing ICI. Some steps have been taken in the field of ICI-induced colitis, but many questions remain in other areas including rheumatic irAEs. In this context, we present our preliminary experience with a selected group of patients with arthritis induced by ICI who had even higher serum calprotectin levels than patients with active conventional inflammatory arthritis, such as RA. Validation and prospective data in other populations are needed; however, we propose the use of serum calprotectin as a useful biomarker in patients with rheumatic (mainly arthritis) irAEs in conjunction with clinical and conventional inflammatory markers.

We are aware that the OMERACT irAE working group is dealing with the harmonization of several concepts related to ICI-induced arthritis, including the homogenization of definitions concepts and the inclusion of biomarkers [49].

## Figures and Tables

**Figure 1 cancers-15-02984-f001:**
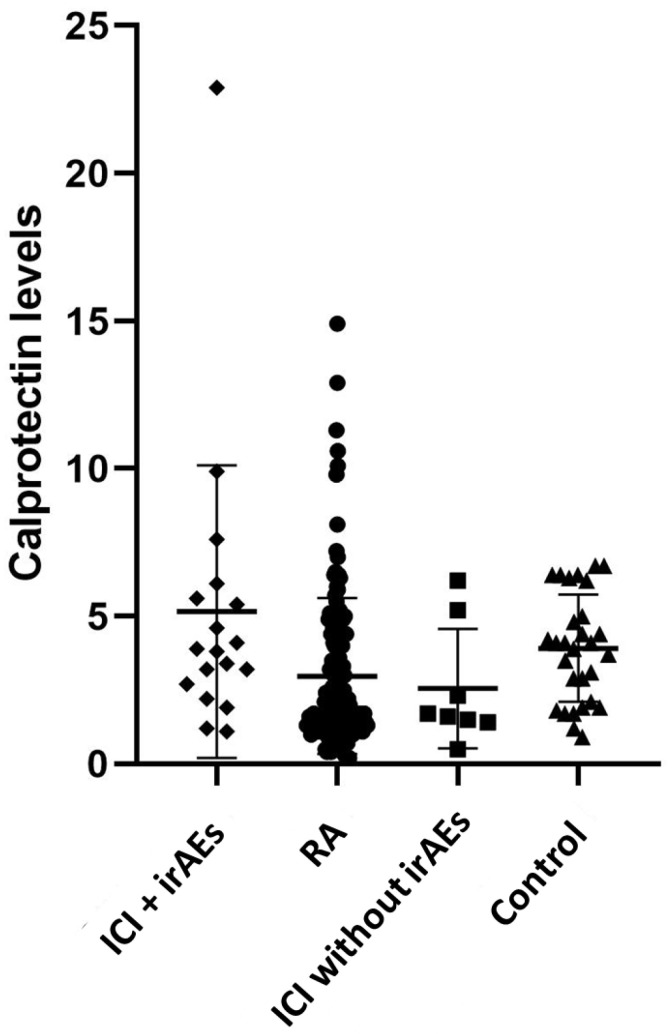
Diagram representing calprotectin levels in the groups. Rhombus represent ICI + irAEs: immune checkpoint inhibitors with rheumatic immune-related adverse event; circles represent RA: rheumatoid arthritis. Square represents ICI without irAEs: immune checkpoint inhibitor control group. Triangle represents control: healthy donors. Calprotectin levels (μg/mL). Statistical significance was reached when comparing the calprotectin levels between the main group (ICI + irAEs) and the control group of oncologic patients without irAEs (*p*-value 0.49, calculated with the Mann–Whitney test).

**Figure 2 cancers-15-02984-f002:**
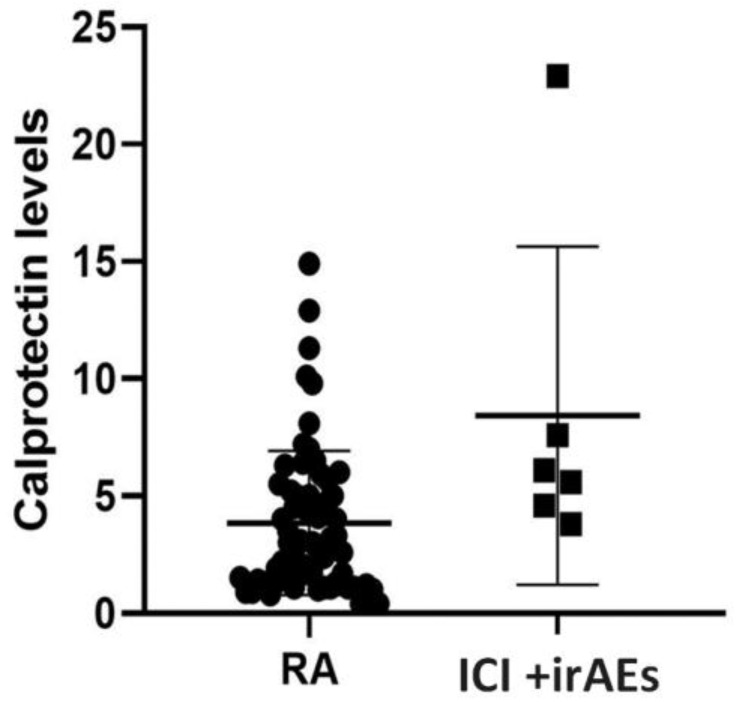
Calprotectin levels in relation to inflammatory arthritis at the time of determination. Circles represent RA: rheumatoid arthritis. Squares represent ICI + irAEs: immune checkpoint inhibitors plus irAEs. Calprotectin levels (μg/mL).

**Figure 3 cancers-15-02984-f003:**
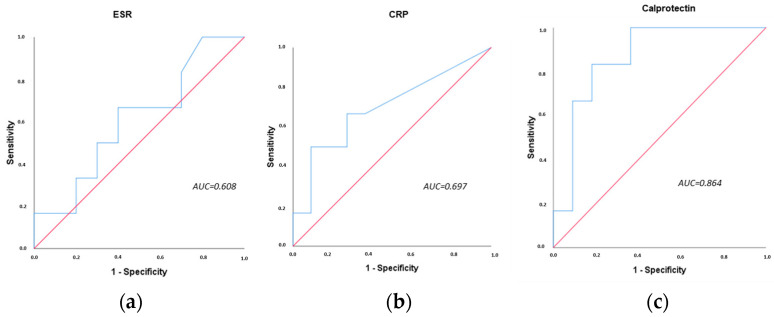
ROC curves of blood biomarkers for inflammatory activity in patients with rheumatic irAEs. (**a**) ROC curve of ESR, (**b**) ROC curve of CRP, (**c**) ROC curve of serum calprotectin. AUC: area under the curve. ESR: erythrocyte sedimentation rate, CRP: C-reactive protein.

**Figure 4 cancers-15-02984-f004:**
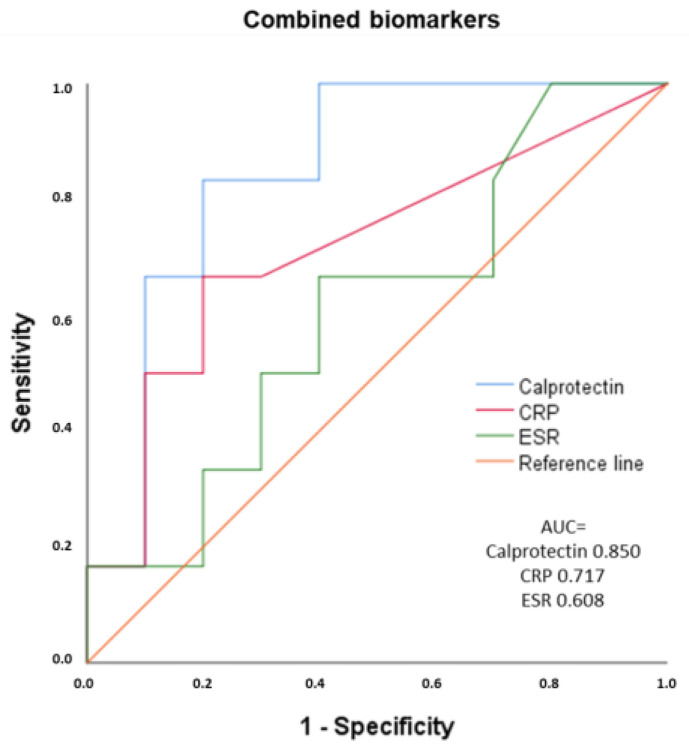
ROC curves of combined calprotectin, CRP, and ESR for inflammatory activity in patients with musculoskeletal irAEs. AUC: area under the curve. ESR: erythrocyte sedimentation rate. CRP: C-reactive protein.

**Table 1 cancers-15-02984-t001:** General characteristics of patients with rheumatic irAEs.

Patient	Type of Neoplasia	Age (Years)	Sex	Pre-Rheum	Previous irAEs	Type of Cancer Therapy	Rheumatic irAE Presentation	Serum Calprotectin (μg/mL)	Active Arthritis	Treatment *
1	Melanoma	58	Male	Cryoglobulinemia	Vitiligo	Nivolumab	Oligo/polyarthritis **	5.4	No	GC
2	Lung (squamous)	65	Female	None	None	Pembrolizumab	RA-like	3.8	Yes	GC
3	Melanoma	65	Male	None	None	Pembrolizumab + Epacadostat	Arthralgia	3.2	No	No
4	Melanoma	68	Male	Fibromyalgia	None	Pembrolizumab + Epacadostat	Oligo/polyarthritis *	22.9	Yes	No
5	Melanoma	53	Male	None	None	Pembrolizumab	Arthralgia	5.6	Yes	No
6	Breast	54	Male	None	None	Atezolizumab	Arthralgia	7.6	Yes	GC
7	Melanoma	76	Female	None	Hypophysitis thyroiditis	Ipilimumab + Nivolumab	Oligo/polyarthritis *	1.9	No	No
8	Urothelial bladder	61	Female	None	None	Durvalumab	RA-like	3.2	No	Methotrexate
9	Melanoma	59	Female	None	Sarcoidosis	Nivolumab	PMR-like	1.1	No	No
10	Lung	61	Female	Seronegative arthritis	None	Pembrolizumab	Arthralgia	2.7	No	GC
11	Ovarium	80	Male	RA	None	Pembrolizumab	RA-like	4.6	Yes	Methotrexate and GC
12	Thyroid	73	Male	None	None	Durvalumab + Tremelimumab	PM-like	4.1	No	GC
13	Renal	77	Male	None	Colitis, hepatitis, hypothyroidism	Ipilimumab+ Nivolumab	PMR-like	3.9	No	Methotrexate
14	Lung	63	Female	None	None	Atezolizumab	RA-like	3.4	no	No
15	Lung	48	Male	None	Hypothyroidism	Nivolumab	Oligo/polyarthritis *	1.2	No	GC
16	Liver	55	Female	None	Sensitive motor axonal motor polyneuropathy	Durvalumab	PMR-like	2.2	No	GC
17	Prostate	78	Female	None	None	Nivolumab	Oligo/polyarthritis *	9.9	No	No
18	Melanoma	75	Male	None	None	Nivolumab	Oligo/polyarthritis *	6.1	Yes	GC and Hydroxichloroquine

Pre-rheum: previous rheumatic condition. * Treatment at the time of calprotectin determination. ** Oligo/polyarthritis with or without enthesitis and tenosynovitis. GC: glucocorticosteroids.

**Table 2 cancers-15-02984-t002:** Main characteristics of the control group of patients with rheumatoid arthritis.

	Total (*n* = 128)
Age, mean (sd)	55.92 (11.86)
Female, *n* (%)	117 (90.7)
Disease evolution (years), mean (sd)	15.1 (10.0)
28TJC, mean (sd)	4.43 (6.1)
28SJC, mean (sd)	2.2 (3.0)
PGA, mean (sd)	4.1 (2.7)
PhGA, mean (sd)	3.2 (2.7)
CDAI, mean (sd)	13.9 (12.1)
SDAI, mean (sd)	14.6 (13.5)
DAS28, mean (sd)	3.38 (1.60)

28SJC: 28 swollen joint counts; 28TJC tender joint count; PGA: patient global assessment, PhGA: global assessment; CDAI: clinical disease activity index; SDAI: simplified disease activity index; DAS28: disease activity score. *n*: number of patients; sd: standard deviation.

**Table 3 cancers-15-02984-t003:** Levels of classical acute-phase reactants in the different groups of patients.

	ICI Patients (*n* = 18)	Healthy Donors (*n* = 29)	*p*-Value (ICI vs. Healthy Donors)	RA (*n* = 128)	*p*-Value (ICI vs. RA)	Total (*n* = 175)
ESR mm/h. mean (sd)	28.50 (34.15)	8.10 (4.57)	0.004	16.06 (19.27)	0.05	16.56 (21.0)
CRP mg/dL, mean (sd)	3.31 (6.28)	0.87 (0.18)	<0.001	0.86 (1.60)	0.01	1.10 (2.63)

ICI: immune checkpoint inhibitors. ESR: erythrocyte sedimentation rate, CRP: C-reactive protein. Sd: standard deviation. U Mann–Whitney test.

**Table 4 cancers-15-02984-t004:** Comparison of classic acute-phase reactants in ICI patients with active inflammatory activity and patients with active RA (DAS28 > 2.6).

	ICI Patients (*n* = 6)	RA (*n* = 79)	*p* Value
ESR mm/h, mean (sd)	39.33 (50.0)	21.78 (22.6)	0.4
CRP mg/dL, mean (sd)	5.17 (9.5)	1.17 (2.0)	0.08

ICI: immune checkpoint inhibitors. ESR: erythrocyte sedimentation rate. CRP: C-reactive protein. Sd: standard deviation. U Mann–Whitney.

**Table 5 cancers-15-02984-t005:** Correlation between inflammatory biomarkers in the group of patients with irAEs.

	ESR	CRP	Serum Calprotectin
ESR	1	0.434 **	0.283 **
CRP	0.434 **	1	0.432 **
Serum calprotectin	0.283 **	0.432 **	1

** *p*-Value < 0.01. Spearman’s correlation coefficient. ESR: erythrocyte sedimentation rate. CRP: C-reactive protein [26,27,28,29,30,31,32,33,34,35,36,37,38,39,40].

## Data Availability

Not applicable.

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
