# Peer review of "Calprotectin in Patients with Rheumatic Immunomediated Adverse Effects Induced by Checkpoints Inhibitors"

_cancers, 2023, doi:10.3390/cancers15112984_

Round 1

Reviewer 1 Report (Previous Reviewer 1)

I value the authors' effort to improve their work, primarily recruiting a new control group of oncologic patients under ICI but with no IRAEs. I suggest adding their results to Figure 1 (diagram). In this figure, the reference group (ICI+IRAE) should be placed as the first plot (left), and then the other control groups (RA, ICI-IRAE neg, healthy controls). I also suggest plotting the p-value for comparisons between cases and each control group.

For CRP/ESR, are these data also available from ICI-irAE-negative patients? If so, I would include them in the manuscript and Table 3. Here, I don't entirely agree with the statistical analyses performed. The authors conducted an ANOVA test between the three groups (by the way, was the data normality checked?), but this test allows to detection of global differences but with no specification. Indeed, the significant value could be due to comparing RA and healthy controls. Similarly to calprotectin levels, the authors should provide individual comparisons between the group of interest and each control group, using Mann-Whitney or t-test, as appropriate.

Regarding ICI-irAEs patients, I miss data about treatment for the rheumatic event (GC, DMARD, biologics) at the time of the calprotectin assessment. Please include it, at least in Table 1.

The source of healthy controls and oncologic controls should be stated in Methods.

From my point of view, section 3.5 (Calprotectin in other irAEs), mainly focused on colitis, does not fit in the paper, primarily focused on rheumatic irAEs. I suggest the authors write a summary paragraph and place it in the Introduction.

Author Response

# Reviewer 1

I value the authors' effort to improve their work, primarily recruiting a new control group of oncologic patients under ICI but with no IRAEs. I suggest adding their results to Figure 1 (diagram). In this figure, the reference group (ICI+IRAE) should be placed as the first plot (left), and then the other control groups (RA, ICI-IRAE neg, healthy controls). I also suggest plotting the p-value for comparisons between cases and each control group.

  • Thank you to the reviewer for their comments. Regarding including Figure 1, according to your recommendations, we have modified the graph. We have included in the text the p-value when comparing the calprotectin levels of the main group with the new control group of patients treated with ICI but without IRAEs.

For CRP/ESR, are these data also available from ICI-irAE-negative patients? If so, I would include them in the manuscript and Table 3.

  • Unfortunately, we do not have the values of CRP and ESR at the time of calprotectin determination for these patients, as it was a specific determination that was requested by the reviewer to include as a control group for this study. We have reviewed the electronic medical records of these patients again to verify if there were any CRP or ESR tests conducted around the time of the calprotectin determination. We have noticed that most of these tests are separated by more than 4 months.

Here, I don't entirely agree with the statistical analyses performed. The authors conducted an ANOVA test between the three groups (by the way, was the data normality checked?), but this test allows to detection of global differences but with no specification. Indeed, the significant value could be due to comparing RA and healthy controls. Similarly to calprotectin levels, the authors should provide individual comparisons between the group of interest and each control group, using Mann-Whitney or t-test, as appropriate.

  • Thank you to the reviewer for your comments. We have reviewed the distribution of calprotectin levels and found that it was not normal. Following the reviewer's recommendations, we have revised the statistics of the study. We have compared the calprotectin levels between the main group and each control group using the Mann-Whitney test, and we have included the results in the text.

We now stated:

  • (line 178) “The mean serum calprotectin levels in patients with rheumatic irAEs were 5.15 + 4.96 μg/ml. In the control group of RA patients, the mean serum calprotectin was 3.19 + 3.60 μg/ml. This difference did not reach statistical significance (p value=0.8). In the healthy donor group the mean serum calprotectin was 3.81 + 1,86μg/ml, significantly higher in comparison with the main group (p value=0.006). Furthermore, we performed a determination of serum calprotectin in 8 patients receiving ICI treatment but without irAEs who volunteered for the study. The mean levels of calprotectin in these patients was 2.41 + 2.05 μg/ml (Figure 1). We found that these levels were significantly lower than those of patients with irAEs (p value=0.49) and without significant difference compared to the healthy controls (p value 0.56)”.
  • Table 3: Levels of classical acute phase reactants in the different groups of patients.

ICI patients (N=18)

Healthy donors (N=29)

p-value

(ICI vs Healthy donors)

RA (N=128)

p-value

(ICI vs RA)

TOTAL (N=175)

ESR mm/h. mean (sd)

28.50 (34.15)

8.10 (4.57)

0.004

16.06 (19.27)

0.05

16.56 (21.0)

CRP mg/dL, mean (sd)

3.31 (6.28)

0.87 (0.18)

<0.001

0.86 (1.60)

0.01

1.10     2.63)

* ICI: immune checkpoint inhibitors. ESR: erythrocyte sedimentation rate, CRP: C reactive protein. Sd: standard deviation. U Mann-Whitney test.

  • line 206: When classifying patients with ICI as patients with active inflammatory activity (n=6) versus patients without inflammatory activity (n=12) at the time of the calprotectin assessment, we observed that serum calprotectin levels were higher in the group of active patients (8.43μg/ml (7.2) vs. 3.53μg/ml (2.47)). Our findings showed a statistical trend toward higher levels of serum calprotectin levels in patients with inflammatory activity (p<0.05). Similar to this, when comparing serum calprotectin levels in the group of patients with active RA (DAS28 >2.6) versus patients in remission, we also found significant differences in serum calprotectin levels in active RA patients (3.94μg/ml (4.25) vs. 2.03μg/ml (1.72), p<0.05)).

3.3 Serum calprotectin levels in patients with active arthritis

We performed a sub-analysis to compare the serum levels of calprotectin in patients with active arthritis of both groups (6 patients in the study group and 79 in the RA control group). Although both groups had elevated serum calprotectin levels, we found significantly higher levels of serum calprotectin in the group of patients treated with ICI (8.43 (7.2) μg/ml vs 3.94 (4.25), p=0.013)) (Figure 2). No significant differences were observed with CRP or ESR (Table 4).

Regarding ICI-irAEs patients, I miss data about treatment for the rheumatic event (GC, DMARD, biologics) at the time of the calprotectin assessment. Please include it, at least in Table 1.

  • Thank you for pointing that out. We have included the data in Table 1.

Table 1. General characteristics of patients with rheumatic irAEs

Patient

Type of neoplasia

Age (yrs)

Sex

Pre-rheum

Previous

irAEs

Type of cancer therapy

Rheumatic irAE presentation

Serum calprotectin ( μg/ml)

Active arthritis

Treatment*

1

Melanoma

58

Male

Cryoglobulinemia

Vitiligo

Nivolumab

Oligo/polyarthritis**

5,4

No

GC

2

Lung (squamous)

65

Female

None

None

Pembrolizumab

RA-like

3,8

Yes

GC

3

Melanoma

65

Male

None

None

Pembrolizumab +

Epacadostat

Arthralgia

3,2

No

No

4

Melanoma

68

Male

Fibromyalgia

None

Pembrolizumab + Epacadostat

Oligo/polyarthritis*

22,9

Yes

No

5

Melanoma

53

Male

None

None

Pembrolizumab

Arthralgia

5,6

Yes

No

6

Breast

54

Male

None

None

Atezolizumab

Arthralgia

7,6

Yes

GC

7

Melanoma

76

Female

None

Hypophysitis thyroiditis

Ipilimumab +Nivolumab

Oligo/polyarthritis*

1,9

No

No

8

Urothelial bladder

61

Female

None

None

Durvalumab

RA-like

3,2

No

Methotrexate

9

Melanoma

59

Female

None

Sarcoidosis

Nivolumab

PMR-like

1,1

No

No

10

Lung

61

Female

Seronegative arthritis

None

Pembrolizumab

Arthralgia

2,7

No

GC

11

Ovarium

80

Male

RA

None

Pembrolizumab

RA-like

4,6

Yes

Methotrexate and GC

12

Thyroid

73

Male

None

None

Durvalumab + Tremelimumab

PM-like

4,1

No

GC

13

Renal

77

Male

None

Colitis,

hepatitis,

hypothyroidism

Ipilimumab+ Nivolumab

PMR-like

3,9

No

Methotrexate

14

Lung

63

Female

None

None

Atezolizumab

RA-like

3,4

no

No

15

Lung

48

Male

None

Hypothyroidism

Nivolumab

Oligo/polyarthritis*

1,2

No

GC

16

Liver

55

Female

None

Sensitive motor axonal motor polyneuropathy

Durvalumab

PMR-like

2,2

No

GC

17

Prostate

78

Female

None

None

Nivolumab

Oligo/polyarthritis*

9,9

No

No

18

Melanoma

75

Male

None

None

Nivolumab

Oligo/polyarthritis*

6,1

Yes

GC and Hydroxichloroquine

Pre-rheum: Previous rheumatic condition. * Treatment at the time of calprotectin determination.

** Oligo/polyarthritis with or without enthesitis and tenosynovitis. GC: glucocorticosteroids.

The source of healthy controls and oncologic controls should be stated in Methods.

  • Thank you for pointing that out. We have included the source of the controls in the Methods section. We now stated: “We also included a second control group of healthy volunteers selected from hospital workers with no medical history of interest. To validate our observations, we also determined the levels of calprotectin in a group of cancer patients undergoing treatment with ICI without any irEAs under follow-up by the oncology service of our hospital who were attending their control visit and who voluntarily agreed to participate in the study as a control group.”

From my point of view, section 3.5 (Calprotectin in other irAEs), mainly focused on colitis, does not fit in the paper, primarily focused on rheumatic irAEs. I suggest the authors write a summary paragraph and place it in the Introduction.

  • As suggested by the reviewer, we have removed section 3.5 from the results.

Reviewer 2 Report (Previous Reviewer 3)

The authors addressed most of my concerns properly except for the one below. 

It has been reported that calprotectin can be produced by several immune cells, including plasmacytoid dendritic cells and platelets, while the majority of the calprotectin is derived from neutrophils. Given that the fragile neutrophils undergo spontaneous NET formation upon serum processing, e.g. coagulation, during which calprotectin may be released, and serum calprotectin levels may not reflect their true physiological levels experienced by patients. The authors measured calprotectin levels in the serum. Did the authors also measure their levels in the plasma?

Author Response

We appreciate the reviewer for the new comment. In this case, we have not measured calprotectin in plasma, only in serum, as it is the established technique in our center. This method has been previously studied (Macias-Muñoz L, Frade-Sosa B, Iniciarte-Mundo J, Hidalgo S, Morla RM, Gallegos Y, Sanmarti R, Auge JM. Analytical and clinical evaluation of DiaSorin Liaison® Calprotectin fecal assay adapted for serum samples. J Clin Lab Anal. 2022 Mar;36(3):e24258. doi: 10.1002/jcla.24258.). Calprotectin levels were not studied in plasma in these patients since it is not the standardized technique in our centre.

Round 2

Reviewer 1 Report (Previous Reviewer 1)

Thanks for addressing all my comments; I am mostly satisfied. I have two minor comments:

- In the modified manuscript, I now see two Figures 1? I guess the upper one (old Figure 1) should be removed during the editorial process

- Page 6, lines 196-197. The authors stated, "In the healthy donor group, the mean, serum calprotectin was 3.81 +1,86μg/ml, significantly higher in comparison with the main group (p value=0.006)". This is confusing. Did you mean "lower" compared to the rheumatic irAE group? Because in them, the mean level was 5.15. Could you clarify this point?

Author Response

Thanks for addressing all my comments; I am mostly satisfied. I have two minor comments:

- In the modified manuscript, I now see two Figures 1? I guess the upper one (old Figure 1) should be removed during the editorial process

Thank you for pointing that out. The previous Figure 1 had not been removed properly. We have now corrected it.

- Page 6, lines 196-197. The authors stated, "In the healthy donor group, the mean, serum calprotectin was 3.81 +1,86μg/ml, significantly higher in comparison with the main group (p value=0.006)". This is confusing. Did you mean "lower" compared to the rheumatic irAE group? Because in them, the mean level was 5.15. Could you clarify this point?

Thank you for pointing that out. We have now corrected it. Indeed, it is significantly lower.

This manuscript is a resubmission of an earlier submission. The following is a list of the peer review reports and author responses from that submission.

Round 1

Reviewer 1 Report

I read the paper with interest. The authors present a study comparing serum calprotectine levels among patients with rheumatic events of immune checkpoint inhibitors (ICIs), rheumatoid arthritis, and healthy donors. Soundly, higher levels are noted in those with rheumatic complications of ICIs, but I see a relevant bias in the selection of controls. The suitable control group here should be oncological patients under ICIs but with non-rheumatic complications or with none of them, to be used as potential diagnostic and follow-up marker  - it is my experience that several oncologic patients under ICIs show persistently raised serum inflammatory markers.

In addition, I don't find appropriate accompanying a case-control study with a narrative literature review on serum calprotectine.

Reviewer 2 Report

intersting well organised scientific work, prototypical, that may indicate a potential diagnostic tool for ICI induced muscoskeletal manifestations.

Although ICI discontinuation is performed after severe clinical aggravation, calprotectin levels may indicate earlier discontinuation.

Reviewer 3 Report

In this paper, the authors reported that serum levels of calprotectin were higher in patients with active rheumatic irAEs when compared to the controls. The authors also conducted a brief literature review on the potential use of calprotectin in colitis, a common gastrointestinal irAEs. These data add to the current literature suggesting that serum calprotectin may serve as a biomarker of inflammatory activity in patients with musculoskeletal irAEs induced by treatment with immune checkpoint inhibitors (ICI).  There are several concerns listed below that would strengthen the paper.

1.       It has been reported that calprotectin can be produced by several immune cells, including plasmacytoid dendritic cells and platelets, while the majority of the calprotectin is derived from neutrophils. Given that the fragile neutrophils undergo spontaneous NET formation upon serum processing, e.g. coagulation, during which calprotectin may be released, and serum calprotectin levels may not reflect their true physiological levels experienced by patients. The authors measured calprotectin levels in the serum. Did the authors also measure their levels in the plasma?

2.       Besides colitis, what about the levels of calprotectin in other common irAEs?

3.       Table 3: what types of statistical analysis did the authors use? ANOVA or Mann-Whitney U-tests? There was only one p-value reported in the table regarding each biomarkers, however, it was described in the legend that Mann-Whitney U-tests were used? Please clarify.

4.       There is no need to present the data in formats of both table and figure. For example, Table 3 and Figure 1, please keep one format. I believe Table 4 and Figure2 was the same situation.

5.       In the discussion, line 363-365, the authors stated that they found higher levels of conventional biomarkers such as ESR and CRP in patients with rheumatic irAEs in comparison with RA patients and healthy controls as well as higher levels of serum calprotectin. Based on the Table 3, the levels of ESR and CRP, but not calprotectin, in patients with rheumatic irAEs were higher than controls. Please correct.

6.       There were some typos.

a.       Line 182: the calprotectin levels in ICI patients were 5.15 + 5.96 μg/ml, while in table 3, they were reported as 5.15 (4.96). Please correct.

b.       Line 257: r=0.0283. Should it be r=0.283?

c.        Line 259: r= 0.434. r was left out.